# Exploring the Water–Soil–Crop Dynamic Process and Water Use Efficiency of Typical Irrigation Units in the Agro-Pastoral Ecotone of Northern China

**DOI:** 10.3390/plants13141916

**Published:** 2024-07-11

**Authors:** Guoshuai Wang, Xiangyang Miao, Bing Xu, Delong Tian, Jie Ren, Zekun Li, Ruiping Li, Hexiang Zheng, Jun Wang, Pengcheng Tang, Yayang Feng, Jie Zhou, Zhiwei Xu

**Affiliations:** 1Yinshanbeilu Grassland Eco-Hydrology National Observation and Research Station, China Institute of Water Resources and Hydropower Research, Beijing 100038, China; imau_wgs@163.com (G.W.); renj@iwhr.com (J.R.); tangpc1988@163.com (P.T.);; 2Institute of Water Resources for Pastoral Area, Ministry of Water Resources, Hohhot 010020, China; 3College of Water Conservancy and Civil Engineering, Inner Mongolia Agricultural University, Hohhot 010018, China; nmglrp@imau.edu.cn; 4Farmland Irrigation Research Institute, Chinese Academy of Agricultural Sciences (CAAS), Xinxiang 453002, China; 5Agriculture, Animal Husbandry and Water Resources Bureau of Saihan District, Hohhot 010018, China; xly001225@163.com

**Keywords:** agro-pastoral ecotone of Northern China, the different forage crops, DSSAT model, soil water dynamics, crop growth processes, crop evapotranspiration, water productivity

## Abstract

Groundwater resources serve as the primary source of water in the agro-pastoral ecotone of northern China, where scarcity of water resources constrains the development of agriculture and animal husbandry. As a typical rainfed agricultural area, the agro-pastoral ecotone in Inner Mongolia is entirely dependent on groundwater for agricultural irrigation. Due to the substantial groundwater consumption of irrigated farmland, groundwater levels have been progressively declining. To obtain a sustainable irrigation pattern that significantly conserves water, this study faces the challenge of unclear water transport relationships among water, soil, and crops, undefined water cycle mechanism in typical irrigation units, and water use efficiency, which was not assessed. Therefore, this paper, based on in situ experimental observations and daily meteorological data in 2022–2023, utilized the DSSAT model to explore the growth processes of *potato*, *oat*, *alfalfa*, and *sunflower*, the soil water dynamics, the water balance, and water use efficiency, analyzed over a typical irrigation area. The results indicated that the simulation accuracy of the DSSAT model was ARE < 10%, nRMSE/% < 10%, and R^2^ ≥ 0.85. The consumption of the soil moisture during the rapid growth stage for the *potatoes*, *oats*, *alfalfa*, and *sunflower* was 7–13% more than that during the other periods, and the yield was 67,170, 3345, 6529, and 4020 kg/ha, respectively. The soil evaporation of oat, potato, alfalfa, and sunflower accounted for 18–22%, 78–82%; 57–68%, and 32–43%, and transpiration accounted for 40–44%, 56–60%, 45–47%, and 53–55% of ETa (333.8 mm–369.2 mm, 375.2 mm–414.2 mm, 415.7 mm–453.7 mm, and 355.0 mm–385.6 mm), respectively. It was advised that irrigation water could be appropriately reduced to decrease ineffective water consumption. The water use efficiency and irrigation water use efficiency for *potatoes* was at the maximum amount, ranging from 16.22 to 16.62 kg/m^3^ and 8.61 to 10.81 kg/m^3^, respectively, followed by alfalfa, sunflowers, and oats. For the perspective of water productivity, it was recommended that *potatoes* could be extensively cultivated, alfalfa planted appropriately, and oats and sunflowers planted less. The findings of this study provided a theoretical basis for efficient water resource use in the agro-pastoral ecotone of Northern China.

## 1. Introduction

The Yinshanbeilu region, which stretches across the central part of Inner Mongolia, represents a typical agro-pastoral ecotone in the north of China. This area serves dual functions in production and ecology; however, agriculture and animal husbandry are constrained by insufficient water resources. The area generally supports the cultivation of a short-growing period, cold- and drought-resistant crops such as *oats*, *potatoes*, and *sunflowers*, and forage crops [1]. Soil erosion is severe in this region and the long-term cultivation in grasslands and improper planting practices have led to coarsening of the soil texture [2,3]; therefore, the soil water retention and fertilizer capacity have drastically declined [4,5]. The local agricultural development is facing serious threats. The region experiences a dry climate. With the increase in crop varieties and irrigation, the over-extraction of groundwater has become serious, which has led to water scarcity becoming a primary limiting factor for agricultural and pastoral production [6,7]. In the agro-pastoral ecotone of northern China, agricultural irrigation relies entirely on groundwater. The higher water consumption characteristics of crops, such as *potatoes*, *sunflowers*, and forage crops [8,9,10], mean their cultivation is constrained by local climate and water resources. Establishing rational planting patterns [11] and optimizing irrigation systems [12] can fully utilize space, improve soil structure, and increase water productivity and water use efficiency; however, the region currently faces challenges, including the presence of unclear relationships in water transport among soil, crops, and irrigation, and undefined water cycle processes and water use efficiency. Thus, it is important to explore the dynamic processes of water–soil–crops and subsequently improve water use efficiency and yield. Consequently, numerous scholars have researched the water resource utilization and crop yield of the agro-pastoral ecotone. Wang et al. [13], based on economic benefits, tuber quality, and soil environmental benefits, applied the Analytic Hierarchy Process and Fuzzy Comprehensive Evaluation Method to optimize irrigation and fertilization and found that the optimal pattern for irrigation is 100% ET_C_ and for fertilization (N-P-K) is 200-34.9-248.9 kg/ha. To increase oat yields, Ma et al. [14], based on the APSIM model, studied the yield and water–nitrogen response under rainfed and irrigated conditions and found that it was recommended to apply nitrogen at 90 kg/ha for rainfed conditions and apply nitrogen at 120 kg/ha with an irrigation quota of 120 mm for irrigation conditions. Liu et al. [15] studied the relationship between water use efficiency and alfalfa yield in different irrigation methods (Flood irrigation (Appendix A), FI, and subsurface drip irrigation, SDI) and found that water use efficiency in alfalfa production for SDI was 3.0–17.8 kg/(ha·mm) higher than that for FI. Zhang et al. [16] used the Two-Factor Design Method to study the optimal planting methods and conservation tillage pattern for sunflower in arid regions. Extensive research had been conducted on the crop physiology, irrigation methods, and irrigation scheduling of *potatoes*, *oats*, *alfalfa*, and *sunflowers*. However, currently, research is mainly focused on study areas with flat terrains, long growing seasons, and higher accumulated temperatures. In contrast, there is relatively less research on crops in agro-pastoral ecotones with shorter growing periods and lower accumulated temperatures, and few studies have researched the dynamic processes of water–soil–crops, water balance or water use efficiency in agro-pastoral ecotones.

Crop growth models can quantitatively demonstrate the process of crop growth, dynamics of crop yield, and the impact of the environment on these factors [17]. These multifunctional, spatial, digital, and visual models enable the more accurate prediction of crop growth patterns, potential, and climate effects [18,19]. The Decision Support System for Agrotechnology Transfer (DSSAT) is one of the most widely used crop growth models worldwide [20,21]. Miao et al. [22] employed the DSSAT model to study the growth process of perennial legume alfalfa, revealing the feedback relationship between alfalfa yield and soil moisture. Furthermore, they identified the optimal soil moisture levels for alfalfa regreening as well as the best irrigation regime for this plant. Zhou et al. [23] integrated remote sensing into the DSSAT model to study surface and root-zone soil moisture, assessing the impact of data assimilation on agricultural drought monitoring. Their research showed that soil moisture estimates could be improved significantly by data assimilation, enhancing drought monitoring by at least one drought level as compared to non-assimilated data. Based on the small-hole effect, Shen et al. [24] developed an estimation method for soil evaporation under plastic film mulch and integrated it into the DSSAT model, addressing the lack of a soil evaporation module for plastic film mulch in DSSAT. Meanwhile, Ma et al. [25]—aiming to rapidly evaluate crop genetic parameters for model applications and improve optimization efficiency and accuracy—coupled the Parameter Estimation (PEST) tool (an independent automatic parameter optimization tool) with the DSSAT crop growth model using the R programming language and established the DSSAT-PEST optimization method. Further, Wang et al. [26] assessed various drip irrigation and fertilization schemes for *potato* farming using the DSSAT-SUBSTOR–potato model. Accordingly, they recommended an optimal irrigation level of 100% ETc, with a fertilization rate (N-P_2_O_5_-K_2_O) of 200-80-300 kg/ha, for *potato* farming in the northwest region. Together, these studies demonstrate that the DSSAT model can effectively evaluate agricultural production and help in enhancing the yield of various crops, but the effective use of the model depends on the results of the validation in each zone and with the crops of that zone. However, current research on typical crop irrigation methods and planting patterns in agro-pastoral zones mainly relies on field experiments, and the application of crop growth models in this context has thus far been limited. However, field irrigation experiments are influenced by many factors. Therefore, coupling field trials with DSSAT models can reduce labor and resource utilization, thus improving efficiency. Nevertheless, the success of DSSAT model-based simulations is highly dependent on the appropriate selection of crop genetic parameters, field management parameters, and soil parameters. Hence, it is necessary to optimize the crop genetic parameters for common plants such as *potatoes*, oats, sunflowers, and alfalfa in agro-pastoral zones and propose DSSAT model parameters suitable for irrigated farmlands in these zones.

Therefore, this study utilizes the DSSAT model to investigate the following: (1) soil water dynamics and growth processes for *potatoes*, *oats*, *alfalfa* and *sunflowers*; (2) water balance analysis; and (3) water use efficiency evaluation. This research aimed to provide a theoretical basis for the efficient utilization of agricultural water resources, scientific enhancement of crop yields, and rational optimization of cropping pattern in the agro-pastoral ecotone of northern China.

## 2. Materials and Methods

### 2.1. Study Area Overview

The study area was located in Wuchuan County (Yinshanbeilu agro-pastoral ecotone in Inner Mongolia: 41°08.344′ N, 111°17.580′ E; Altitude: 1648.5 m). Wuchuan County has a total cultivated land area of 144,667 ha, including 28,667 ha of irrigated land. The agricultural irrigation water was groundwater. Due to groundwater over-extraction, the groundwater table had been declining at a rate of 27.50 cm/a to 47.50 cm/a, with the depth of wells increasing by 50 to 120 m over the past two decades. The climate of the study area is characterized by a temperate continental monsoon climate with an average annual temperature of 2.5 °C, annual precipitation ranging from 100 to 200 mm, and a frost-free period of 90 to 120 days. The bulk density of soil was from 1.5 to 1.7 g/cm^3^, and the soil texture is sandy loam. The main crops were *potatoes*, *oats*, *sunflowers*, and *alfalfa*. Detailed results are presented in Figure 1.

### 2.2. Experimental Design

For this study, we selected a typical irrigation unit within the research area, where we established one groundwater irrigation well. This well serviced a total irrigation area of 35.3 hm^2^, and the area of cultivation encompassed *potatoes* (15.2 hm^2^), *oats* (10.8 hm^2^), *alfalfa* (4.9 hm^2^), and *sunflowers* (4.4 hm^2^). The research site featured a land gradient, with the elevation decreasing by 39 m from the east to the west. We strategically positioned 68 soil sampling points across the research area, with a 100 m interval maintained between each point (Figure 2). Monitoring was performed from 1 May to 30 September, and soil samples were collected on the 1st and 15th of each month in 2022 and 2023. The sampling frequency was increased during the irrigation season. Taking into account crop traits and root development (Appendix A), soil samples were obtained from a depth of 100 cm, and growth indicators such as plant height and leaf area were measured for the *potato*, *oat*, *alfalfa*, and *sunflower* plants. The research area’s initial planting, field management, and irrigation strategies are based on the real local circumstances. Drip irrigation was the chosen method for irrigation within the research area. We meticulously recorded data for each crop, including its growth stages, irrigation schedules, frequency of irrigation, and the allotted irrigation quota.

### 2.3. Test Indicators and Methods

#### 2.3.1. Physical Properties of Soil in the Study Area

We installed smart soil moisture monitoring devices to track soil moisture dynamics for each crop. These sensors were embedded at depths of 10, 20, 30, 40, 50, 60, 70, 80, 90, and 100 cm, and the data collection interval was set to 1 h. Additionally, we used the oven-drying method to determine the soil moisture content across each crop growth stage (there are four phases of growth: early, fast, medium, and late), as well as before and after irrigation to calibrate the smart soil moisture monitoring device data. As a layer, samples are obtained every 20 cm. Soil samples were manually collected from depths of 0–20 cm, 20–40 cm, 40–60 cm, 60–80 cm, and 80–100 cm, with three samples obtained from each layer. The core cutter method was employed to measure soil bulk density, field capacity, and saturated moisture content. We also analyzed the soil samples to determine the levels of available phosphorus, potassium, and organic matter content, thereby determining the fundamental physical parameters of the soil layers within the experimental area. Detailed results are presented in Table 1, Table 2, Table 3 and Table 4.

#### 2.3.2. Meteorological Data Collection

The meteorological data required for the study were obtained from an automatic weather station located within the experimental area (HOBO-U30, Onset Computer Corp., Bourne, MA, USA). The temperature and amount of rainfall during the crop growth period are shown in Figure 3. The amount of rainfall for the crop growth period in 2022 and 2023 was 213.6 mm and 257.8 mm, with the daily average temperature ranges being 0.4–29.8 °C and 1.3–26.9 °C, respectively.

#### 2.3.3. Crop Growth Periods

Data for the different crops were recorded from the time of seeding. We obtained seeding dates, emergence time, plant heights, leaf areas, and harvest times (Table 5).

#### 2.3.4. Crop Irrigation Regimes

The amount of rainfall during the crop growth period for *oats*, *potatoes*, *alfalfa*, and *sunflowers* was 191.7 mm, 185.3 mm, 213.6 mm, and 198.7 mm in 2022 and 241.1 mm, 236.0 mm, 257.8 mm, and 241.2 mm in 2023, respectively. The irrigation quotas for each growth period over the two years are presented in Table 6.

### 2.4. Crop Growth and Water Consumption Models

The DSSAT model—which is one of the most widely applied crop growth models globally and includes numerous modules such as CROPGRO, CERES, SUBSTOR, CANEGRO, and more—was employed. The model has been shown to simulate the dynamics of a wide variety of crops [27,28].

#### 2.4.1. Soil Water Movement Equation

Figure 4 shows the water transport process module, taking the *potato* as an example. Crop transpiration, soil evaporation, surface runoff, and soil profile drainage make up the outputs of the system, whereas rainfall and irrigation water are the inputs of the system.

The formula used by the DSSAT model to calculate changes in soil water content [29] is as follows:(1)ET(EP+ES)=P+I-∆S-R-D
where ∆*S* represents the change in soil water content; *P* is the rainfall; *I* is the irrigation amount; *EP* is the transpiration; *ES* is the soil evaporation; *R* is the surface runoff; and *D* is the soil profile drainage.

The DSSAT model calculates water stress using the following formula [30]:(2)$$\mathrm{SWD}F_{1}=\frac{WS_{P}}{\mathrm{RWUEP}\times EP_{0}}$$
(3)$$\mathrm{SWD}F_{2}=\frac{WS_{P}}{EP_{0}}$$
where *SWDF*_1_ is the first water stress factor in the model; *WSP* denotes potential root water uptake (mm); *RWUEP* is a species-specific characteristic parameter; *EP*_0_ represents the crop water demand (mm); and *SWDF*_2_ is the second water stress factor in the crop model.

#### 2.4.2. Crop Dry Matter Accumulation Equation

The DSSAT model’s equation for dry matter accumulation is [31]
(4)$$∆\mathrm{TOT}=0.758\times \left(\mathrm{PARCE}\times 10^{-6}\times \mathrm{IPAR}\;-0.004\times \mathrm{TOT}\right)\times \mathrm{SWDF}$$
where ∆*TOT* is the daily increment of crop dry matter (t/ha); *PARCE* is the photosynthetically active radiation conversion efficiency (g/MJ); *IPAR* is the intercepted photosynthetically active radiation (MJ/ha); *TOT* is the total dry matter (t/ha); and *SWDF* is the water stress factor affecting dry matter accumulation.

The calculation formula for *PARCE* in the *DSSAT* model is
(5)$$PARCE={\mathrm{PARCE}}_{\max}\times \left\{1\;-\exp\left[-0.008\times \right.\right.\left.\left.\left(T-8\right)\right]\right\}$$
where *PARCE_max_* is the maximum value for photosynthetically active radiation conversion efficiency (g/MJ), and *T* is the daily average temperature (°C).

### 2.5. Model Development, Calibration, and Validation

#### 2.5.1. Simulation Unit Division

Based on field data (e.g., growth stages, leaf area index [LAI], and yield), the SUBSTOR, CERES, OILCROP, and FORAGES models within DSSAT were employed to simulate the growth process and yield of typical crops like *potatoes*, *oats*, *sunflowers*, and *alfalfa* in the agro-pastoral ecotone of the northern foothills of Yinshan. The simulation periods for *oats*, *potatoes*, *sunflowers*, and *alfalfa* in 2022–2023 were set as 19 May to 1 September, 8 May to 20 September, 29 May to 28 September, and 14 May to 30 September, respectively.

#### 2.5.2. Meteorological Data

The meteorological parameters required for the DSSAT model include solar radiation (MJ·m^−2^), maximum temperature (°C), minimum temperature (°C), and rainfall (mm). The field management parameters required by the model include crop variety, planting method, planting date, planting density and depth, irrigation quota and timing, and fertilization amount and timing, among others. For this model, solar radiation must be calculated based on the duration of sunshine recorded by the weather station, as follows [32]:(6)$$R_{s}=R_{\max}\left(a_{s}-b_{s}\frac{n}{N}\right)$$
where *R_s_* is the total solar radiation (MJ/m^2^); *a_s_* is the clear sky solar radiation constant; *b_s_* is a proportionality factor, both of which are constants related to atmospheric conditions, usually taken as 0.25 and 0.5, respectively; *n* denotes sunshine duration (h); *N* is the maximum possible sunshine duration (h); and *R_max_* is the solar radiation on a clear day (MJ/m^2^).
(7)$$R_{\max}=37.586\times d\times \left(W_{s}+\sin\emptyset \sin\delta +\cos\emptyset \cos\delta \sin w_{s}\right)$$
where R_max_ is the solar radiation on a clear day (MJ/m^2^), *d* is the distance between the earth and the sun, *W_s_* is the sunset hour angle (°), *δ* is the declination of the equator (°), and *∅* is the latitude of the meteorological station (°).
(8)$$N=\frac{24}{\pi }W_{s}$$
where *N* represents the maximum possible duration of daylight (h), and *W_s_* is the sunset hour angle (°).
(9)$$\delta =0.4093\times \sin\left(\frac{2\pi }{365}\times J-1.045\right)$$
where *δ* is the declination of the equator (°), and *J* is the Julian day, which is the day of the year, with 1 January equal to 1.
(10)$$W_{s}=\mathrm{arc}\cos\left(-\tan\emptyset \tan\delta \right)$$
where *Ws* is the sunset hour angle (°), *δ* is the declination of the equator (°), and *∅* is the latitude of the meteorological station (°).

#### 2.5.3. Soil Parameters

The soil parameters required for the DSSAT model consist of the basic physicochemical properties of the soil at different depths, including field capacity, saturated moisture content, soil bulk density, soil organic carbon content, total nitrogen content, and pH value, as shown in Table 1, Table 2, Table 3 and Table 4.

#### 2.5.4. Model Calibration and Validation

In this study, the absolute relative error (*ARE*), root mean square error (*RMSE*), normalized root mean square error (*nRMSE*), and coefficient of determination (*R*^2^) [33,34,35] were used to measure the relative difference between simulated values from the DSSAT model and actual measurements. Generally, the closer the *ARE* is to 0, the higher the model’s accuracy. An *nRMSE* < 10% is considered excellent and 10–20% is good. The *R*^2^ is used to test the fit between model simulations and actual measurements, with values closer to 1 indicating a higher degree of fit. The formula for the index calculation is as follows:(11)$$ARE=\frac{\left|S_{i}-M_{i}\right|}{M_{i}}\times 100\%$$
(12)$$MSE=\sqrt{\frac{{\sum_{i=1}^{n}\left(M_{i}-S_{i}\right)}^{2}}{n}}$$
(13)$$\mathrm{nRMSE}=\frac{\mathrm{RMSE}}{M}\times 100\%$$
(14)$$R^{2}=\frac{{\left[\sum_{i=1}^{n}\left(M_{i}-M\right)\left(S_{i}-S\right)\right]}^{2}}{\sum_{i=1}^{n}{\left(M_{i}-M\right)}^{2}\sum_{i=1}^{n}{\left(S_{i}-S\right)}^{2}}$$
where *Mi* is the observed value, *Si* is the simulated value, *M* is the average of the measured values, *S* is the average of the simulated values, and n is the number of samples.

### 2.6. Crop Irrigation Evaluation Indices

This paper uses water use efficiency (WUE, kg/m^3^) and irrigation water use efficiency (IWUE, kg/m^3^) as two indices [36,37] to evaluate and reflect the relationship between crop yield and irrigation water use. The calculation formula [38] is
(15)$$\mathrm{WUE}=\frac{Y}{10\times \mathrm{ET}}$$
where *Y* is the crop yield (kg/ha), *ET* is the actual evapotranspiration during the growth stage (mm), and 10 is a numerical conversion factor.

To study the sensitivity of crops to irrigation water use, the *IWUE* was calculated as follows:(16)$$\mathrm{IWUE}=\frac{Y-Y_{a}}{10\times I}$$
where *Y* is the crop yield (kg/ha), *Y_a_* is the crop yield without irrigation (kg/ha), *I* is the amount of irrigation water supplied during the growth stage (mm), and 10 is a numerical conversion parameter.

## 3. Results

### 3.1. Model Calibration

In this study, calibration was conducted using experimental data collected in 2022, with *ARE*, *nRMSE*, and *R*^2^ serving as the metrics. According to Table 7, the *ARE* for different crop leaf area indices ranged from 3.06% to 4.18%; the *nRMSE* values ranged between 5.32% and 6.46%; and the *R*^2^ values ranged from 0.87 to 0.89. The *ARE* for yield ranged from 3.47% to 5.82%, with *nRMSE* values ranging between 4.51% and 5.88% and *R*^2^ values ranging from 0.87 to 0.90. Table 8 shows that the accuracy for soil moisture content at 0–60 cm was *ARE* ≤ 7%, *nRMSE* ≤ 10%, and *R*^2^ ≥ 0.85. The calibration of crop genetic parameters is shown in Table 9, Table 10, Table 11 and Table 12. The measured values were in good agreement with the simulated values.

### 3.2. Model Validation

After calibration based on the 2022 data, the corrected genetic parameters for different crops (Table 9, Table 10, Table 11 and Table 12) were added to the corresponding modules for validation. The model was then validated based on the actual data from 2023. The validation results, as shown in Figure 5, indicated that the *ARE* for soil moisture in different crops ranged from 4.23% to 6.04%. Meanwhile, the *nRMSE* values ranged from 4.97% to 5.74% and *R*^2^ ranged from 0.901 to 0.912. The *LAI* in different crops was as follows: *ARE*, 2.85% to 5.37%; *nRMSE*, 4.01% to 5.62%; and *R*^2^, 0.911 to 0.919. The yields were as follows: *ARE*, 3.44% to 5.09%; *nRMSE*, 4.29% to 6.05%; and *R*^2^, 0.917 to 0.930. Hence, the model demonstrated good simulation accuracy, and the calibrated and validated models were deemed suitable for use in this study.

### 3.3. Soil Water Dynamics and Growth Processes of the Different Crops

#### 3.3.1. The Soil Water Content Dynamics for *Potatoes*, *Oats*, *Alfalfa*, and *Sunflowers*

Using the calibrated DSSAT model, we simulated and analyzed the impact of irrigation on soil moisture, crop growth, and yield. The changes in soil moisture content for *potatoes*, *oats*, *alfalfa*, and *sunflowers* are illustrated in Figure 6, Figure 7, Figure 8 and Figure 9. The layer of cultivated soil in the study area was rather thin and had a sandy loam texture. Further, the presence of small pebbles increased with the soil depth, leading to poor water retention.

The saturated moisture contents in the 0–20 cm soil layer for the *potato*, *oat*, *alfalfa*, and *sunflower* plots were 0.35 cm^3^/cm^3^, 0.31 cm^3^/cm^3^, 0.31 cm^3^/cm^3^, and 0.32 cm^3^/cm^3^, respectively. Meanwhile, in the 20–60 cm layer, these values were 0.20–0.26 cm^3^/cm^3^, 0.19–0.24 cm^3^/cm^3^, 0.17–0.22 cm^3^/cm^3^, and 0.18–0.25 cm^3^/cm^3^, respectively. After each irrigation event, the soil moisture level quickly dropped from saturation to maximum water holding capacity.

In 2023, the *potatoes* were irrigated eight times at short intervals. During the rapid growth period (45–75 days), soil moisture decreased by about 60–66%, 52–60%, and 42–51% in the 0–20 cm, 20–40 cm, and 40–60 cm layers before the subsequent round of irrigation, respectively. During the mid-growth stage (75–95 days), soil moisture reduction was around 53–59% in the 0–20 cm layer, 43–48% in the 20–40 cm layer, and 35–40% in the 40–60 cm layer. The water consumption during the rapid growth stage was 7–12% more than that during the mid-growth period.

In 2023, the *oats* were irrigated four times, i.e., at low frequency with longer intervals. During the rapid growth period (30–50 days), soil moisture decreased by about 62–68% in the 0–20 cm layer, 44–52% in the 20–40 cm layer, and 41–49% in the 40–60 cm layer. During the mid-growth period (50–70 days), the reduction was about 51–55% in the 0–20 cm layer, 36–44% in the 20–40 cm layer, and 33–42% in the 40–60 cm layer. The water consumption during the rapid growth period was 11–13% more than that during the mid-growth period.

In 2023, the *alfalfas* were irrigated eight times, i.e., frequently and at relatively short intervals. During the rapid growth periods (30–45 days and 92–105 days), soil moisture decreased by about 50–52% in the 0–20 cm layer, 46–48% in the 20–40 cm layer, and 40–45% in the 40–60 cm layer. During the mid-to-late growth stage (45–50 days and 105–117 days), the reduction was about 43–46% in the 0–20 cm layer, 40–44% in the 20–40 cm layer, and 32–35% in the 40–60 cm layer. The water consumption during the rapid growth period was 7–10% higher than that during the mid-to-late growth stage.

In 2023, the *sunflowers* were irrigated five times, i.e., less frequently and at prolonged intervals. During the early growth period (12–48 days), soil moisture in the 0–20 cm layer decreased by about 43–50% after each irrigation; this reduction was 41–48% in the 20–40 cm layer and 39–47% in the 40–60 cm layer. During the rapid growth period (48–67 days), the reduction was about 56–60% in the 0–20 cm layer, 51–55% in the 20–40 cm layer, and 45–50% in the 40–60 cm layer. The water consumption during the initial growth stage was 10–13% lower than that during the rapid growth period.

#### 3.3.2. The LAI Dynamics for Potatoes, Oats, Alfalfa, and Sunflowers

The changes in the LAI for potatoes, oats, alfalfa, and sunflowers are shown in Figure 10. The LAI of these four crops underwent significant changes during different growth stages. The LAI for *potatoes* peaked on the 45th day after planting, corresponding to substantial soil moisture consumption. To meet the growth needs of the *potatoes*, irrigation was performed four times during this stage. Thereafter, the LAI of *potatoes* began to decline. Meanwhile, the LAI for oats peaked on the 75th day after planting, with rapid growth observed between days 30 and 60, leading to a swift decrease in soil moisture and a higher water demand. During this phase, the oats were irrigated four times to meet water requirements. As oats entered the later stages of growth and development, the LAI started to decrease, and the rate of soil moisture decline started to slow down. Alfalfa exhibited multiple peaks in LAI during its growth, with rapid increases observed following each cutting, indicating a significant increase in water demand with the rapid decrease in soil moisture. As alfalfa entered its blooming stage, the rate of soil moisture decline slowed down. The LAI for sunflowers reached the maximum value on the 67th day after planting, corresponding to significant soil moisture consumption. Irrigation was primarily concentrated during this phase.

#### 3.3.3. The Yield Dynamics for Potatoes, Oats, Alfalfa, and Sunflowers

The yield trends for potatoes, oats, alfalfa, and sunflowers are illustrated in Figure 11. A rapid increase in potato yield occurred during the tuber formation and bulking phase (55–105 days), with significant soil moisture consumption observed at 45–90 days. Throughout the growth period, the yield increased at an average rate of 733.7 kg/ha per day, resulting in a final yield of 67,170 kg/ha. The yield of oats showed exponential growth between 75 and 95 days, and the water demand during this stage was high. The average daily yield increase across the entire growth period was 60.3 kg/ha, resulting in a total yield of 3345 kg/ha. As a perennial plant, alfalfa showed rapid increases in yield at 45–60 and 105–120 days. The average daily yield increase was 35.7 kg/ha for the first cutting and 64.2 kg/ha for the second cutting, with an annual yield of 6529 kg/ha. Sunflower yield increased linearly throughout the growth period, with an average daily yield increase of 124.9 kg/ha. This culminated in a total yield of 4020 kg/ha.

### 3.4. Water Balance Analysis

The changes in water content within the 0–60 cm layer of soil were analyzed throughout the growth season based on the simulated soil moisture levels for *potatoes*, *oats*, *alfalfa*, and *sunflowers* in 2022 and 2023, using the water balance principle. The results are presented in Table 13.

For *oats*, in 2022 and 2023, the evapotranspiration (ET) was 333.8 mm and 369.2 mm, respectively, with total percolation amounts of 20.4 mm and 18.6 mm. Soil evaporation and oat transpiration accounted for 22% and 78% of ET in 2022 and 18% and 82% of ET in 2023, respectively.

For *potatoes*, the ET was 375.2 mm and 414.2 mm in 2022 and 2023, respectively, and the total percolation amount was 24.7 mm and 22.7 mm, respectively. Soil evaporation and potato transpiration accounted for 68% and 32% of ET in 2022 and 57% and 43% of ET in 2023, respectively (Appendix A).

For *alfalfa*, the ET in 2022 and 2023 was 415.7 mm and 453.7 mm, and the total percolation amount was 27.3 mm and 24.8 mm, respectively. Soil evaporation and alfalfa transpiration accounted for 44% and 56% of ET in 2022 and 40% and 60% of ET in 2023, respectively.

For *sunflowers*, the ET was 355 mm and 385.6 mm and the total percolation amount was 18.5 mm and 15.9 mm in 2022 and 2023, respectively. Soil evaporation and sunflower transpiration accounted for 45% and 55% of ET in 2022 and 47% and 53% of ET in 2023, respectively. Among the four crops, alfalfa had the highest ET, percolation amount, and total amount of introduced water.

### 3.5. Water Use Efficiency Analysis

Based on the simulated yield of *potatoes*, *oats*, *alfalfa*, and *sunflowers* (2022 and 2023), and by employing the formulae for WUE and IWUE, the relationships between crop yields and irrigation water usage were analyzed. The results, as shown in Table 14, indicated that the WUE for oats was 1.04 kg/m^3^ in 2022 and 1.01 kg/m^3^ in 2023. This value was relatively consistent, suggesting that the unit water productivity for oat crops remained stable. However, the IWUE decreased from 0.64 kg/m^3^ in 2022 to 0.52 kg/m^3^ in 2023, indicating a reduction in the efficiency of irrigation water use and highlighting the potential need to modify the irrigation strategy. For *potatoes*, the WUE decreased from 16.62 kg/m^3^ in 2022 to 16.22 kg/m^3^ in 2023, while the IWUE dropped from 10.81 kg/m^3^ to 8.61 kg/m^3^. Despite the slight decrease in WUE, the reduction in IWUE was quite significant, since precipitation was 213.6 mm in 2022 and 257.8 mm in 2023, and irrigation was 173 mm in 2022 and 156 mm in 2023. Precipitation in 2023 was more than 44.2 mm, and irrigation in 2023 was less than 17 mm. Rainfall contributed significantly to potato yield. The WUE of alfalfa showed a slight increase from 1.17 kg/m^3^ to 1.44 kg/m^3^ between 2022 and 2023, while its IWUE decreased from 0.93 kg/m^3^ to 0.71 kg/m^3^. Sunflowers maintained a relatively stable WUE, only showing a slight decrease from 1.39 kg/m^3^ to 1.34 kg/m^3^, but their IWUE decreased from 0.85 kg/m^3^ to 0.68 kg/m^3^. Nevertheless, when compared with other crops, sunflowers exhibited a relatively minor decline in water use efficiency. Thus, overall, oats and sunflowers exhibited a stable WUE, with a slight downward trend for IWUE. Potatoes showed a minor decrease in WUE but a more noticeable decline in IWUE. This suggests that the irrigation regime was not effectively adapted to the water demands of the crops.

The value of water productivity and irrigation water use efficiency for *potatoes* was at the maximum amount, ranging from 16.22 to 16.62 kg/m^3^ and 8.61 to 10.81 kg/m^3^, respectively, followed by *alfalfa*, *sunflowers*, and *oats*. For the perspective of water productivity, it was recommended that *potatoes* could be extensively cultivated, *alfalfa* planted appropriately, and *oats* and *sunflowers* planted less.

## 4. Discussion

Using field trial data from 2022 and 2023, this study investigated the soil water dynamics and the growth processes for *potatoes*, *oats*, *Alfalfa*, and *Sunflowers* in the agro-pastoral ecotone. The adjusted model aligned well with the measured data (*ARE* < 10%, *nRMSE*/% < 10%, and *R*^2^ ≥ 0.85), which provided effective predictions of crop growth trends, consistent with findings from Wang et al. (2022) [39], Shen et al. (2020) [40], and Jiang et al. (2019) [41]. The present study also demonstrated that soil moisture changes differed across the different layers of soil at a depth of 0–60 cm, which corresponds to the active rootzone (Appendix A). These results are in line with those from Gao et al. (2015) [42] and Guo et al. (2020) [43], who investigated soil moisture content across different soil depths under various conditions of land use. They demonstrated that the water consumption in the topsoil layer (0–20 cm) is significantly higher than that in the deeper layer (20–40 cm) when the crops have a higher water demand. Similarly, our study revealed that the reduction in soil water within the surface layer (0–20 cm) was 6–11% and 15–19% greater than that in the deeper layer (20–60 cm) for *potatoes* in 2022 and 2023, 11–18% and 13–21% higher for *oats*, 3–4% and 7–11% higher for *Alfalfa*, and 2–5% and 6–10% higher for *Sunflowers*. Additionally, the majority of roots in *potato* and *oat* plants are distributed at a depth of 0–40 cm [44,45]. The significant differences in water absorption and utilization among the four crops led to variations in leaf growth, with the peak LAI being 1.9 for *potatoes*; 16 for *oats*; and 2.7 for *Sunflowers*. Meanwhile, *Alfalfa* exhibited multiple LAI peaks (7.5 and 9). The sensitivity to water deficit across various yield accumulation stages differs among different crops [46], and these differences lead to varying rates of yield increase. For example, Song et al. [47] studied the coupling effect of water and nitrogen on *potato* yield, quality, and water productivity under subsurface drip irrigation conditions in the northwest arid area of Weiwu City. They found that P2N2 (soil moisture ratio of 70% and nitrogen application rate of 135 kg/ha) treatment had the best performance in *potato* tuber starch content and yield, with the highest yield of 54,187 kg/ha, and P1N2 (soil moisture ratio of 40% and nitrogen application rate of 135 kg/ha) treatment had the highest water productivity of 12.86 kg/m^3^. This study found that the yield was 62,355–67,170 kg/ha. The *potato* yield in this study is higher, since the nitrogen fertilizer (N) applied in this study was 185 kg/ha, which was 50 kg/ha more than that of Song et al. [47] and the rainfall in Weiwu City was less than 100 mm.

Simulations based on the DSSAT model were conducted to track water consumption changes, ET, and yield for *potatoes*, *oats*, *Alfalfa*, and *Sunflowers* [48]. The model accurately depicted the water requirements throughout the crop growth process, highlighting the critical role of concentrated rainfall and irrigation in supporting crop growth and enhancing soil moisture content in arid regions [49]. This study identified varying water consumption levels at different stages of crop growth. The total water consumption was found to be 375.2–414.2 mm for *potatoes*, 333.8–369.2 mm for *oats*, 415.7–453.7 mm for *Alfalfa*, and 355.0–385.6 mm for *Sunflowers*. The water consumption for *oats* was consistent with the value reported by Hao et al. (2019) [50], whereas the consumption for *potatoes* was higher than the total water consumption of 216–278 mm reported by Chen et al. (2019) [51]. Conversely, the water consumption calculated for *Alfalfa* and *Sunflowers* was lower than the previously reported water consumption of 400–500 mm for *Alfalfa* [52] and 400 mm for *Sunflowers* [53] during the growth season. The lower values observed for *Alfalfa* and *Sunflowers* were likely due to factors such as shorter growth seasons, less rainfall, and insufficient heat accumulation. Additionally, the water percolation for *potatoes*, *oats*, *Alfalfa*, and *Sunflowers* was 20.1–22.7 mm, 15.9–18.6 mm, 22.7–24.8 mm, and 12.5–15.9 mm, respectively. This suggests that agricultural irrigation management practices should focus on controlling leakage during this period to reduce ineffective water consumption. The irrigation water for the four crops could be reduced to potentially decrease ineffective percolation and consumption. The WUE of *Alfalfa* in 2023 was higher than that in 2022, whereas the WUEs for *oats*, *potatoes*, and *Sunflowers* all decreased from 2022 to 2023, indicating the superior water utilization capability of *Alfalfa*.

Overall, the optimal planting patterns for crops under different water and fertilizer combinations will be examined in future studies based on the DSSAT model, providing a scientific basis for sustainable agricultural development in arid regions.

## 5. Conclusions

The consumption of the soil moisture during the rapid growth stage for the *potatoes*, *oats*, *Alfalfa*, and *Sunflower* was 7–12%, 11–13%, 7–10% and 10–13% more than that during the rapid, middle, mid-to-late, and initial growth period, and the yield was 67,170 kg/ha, 3345 kg/ha, 6529 kg/ha, and 4020 kg/ha, respectively. The water consumption of *oats*, *potatoes*, and *Sunflowers* was 333.8 mm–369.2 mm, 375.2 mm–414.2 mm, and 355.0 mm–385.6 mm, respectively. The maximum water consumption of *Alfalfa* was 415.7 mm–453.7 mm. Among the four crops, *Alfalfa* was the highest water consuming crop. The average percolation of the four crops was 12.5 mm–15.9 mm. The value of water productivity and irrigation water use efficiency for *potatoes* was at the maximum value, ranging from 16.22 to 16.62 kg/m^3^ and 8.61 to 10.81 kg/m^3^, respectively, followed by *Alfalfa*, *Sunflowers*, and *oats*. It was advised that irrigation water could be appropriately reduced to decrease ineffective water percolation and consumption. For the perspective of water productivity, it was recommended that *potatoes* could be extensively cultivated, *Alfalfa* planted appropriately, and *oats* and *Sunflowers* planted less.

## Figures and Tables

**Figure 1 plants-13-01916-f001:**
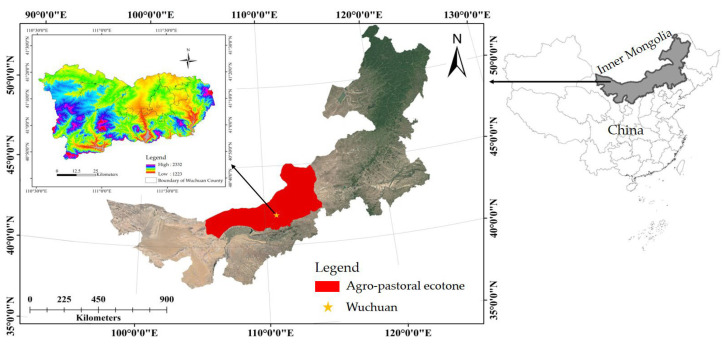
Overview of the study area.

**Figure 2 plants-13-01916-f002:**
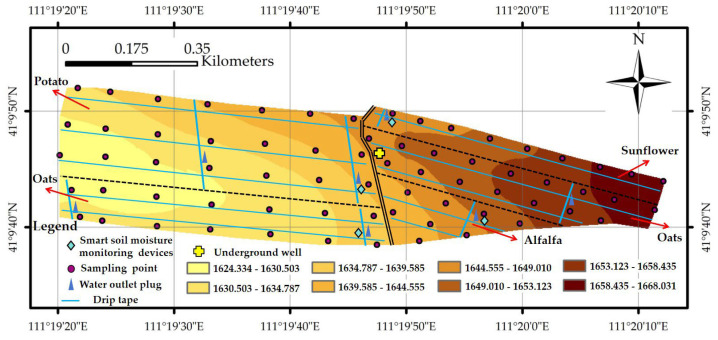
Experimental design diagram.

**Figure 3 plants-13-01916-f003:**
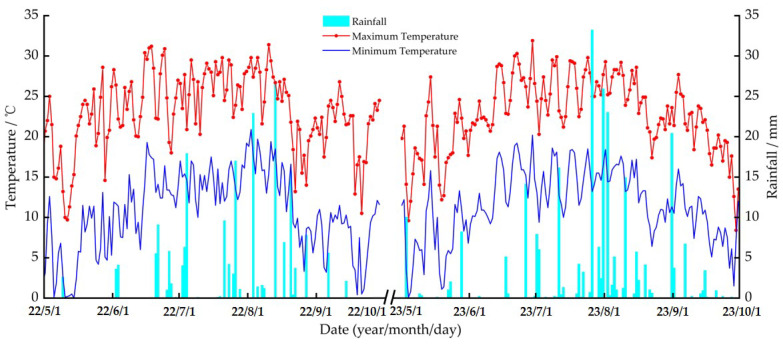
Temperature and rainfall during the growth period.

**Figure 4 plants-13-01916-f004:**
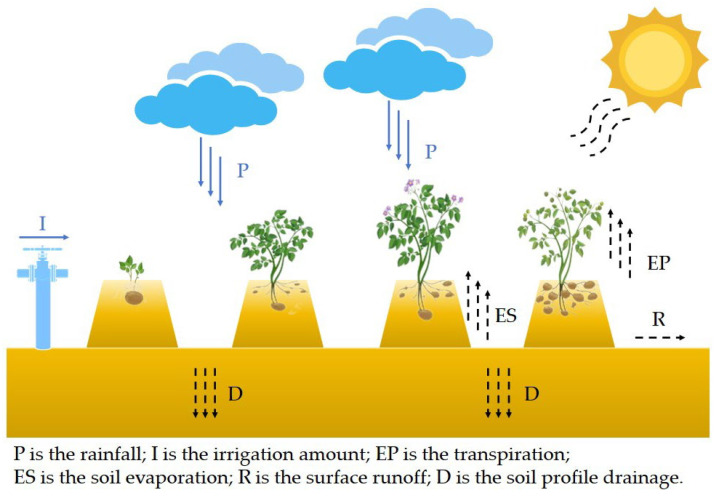
Schematic diagram of water movement process in DSSAT model.

**Figure 5 plants-13-01916-f005:**
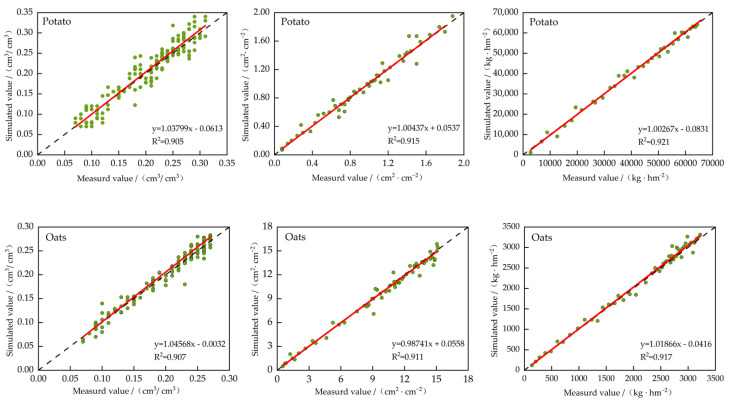
Verification of the soil moisture, leaf area index, and yield in different crops. Note: The green dots are the scatter of the simulated and measured values, and the red line is the fitted line.

**Figure 6 plants-13-01916-f006:**
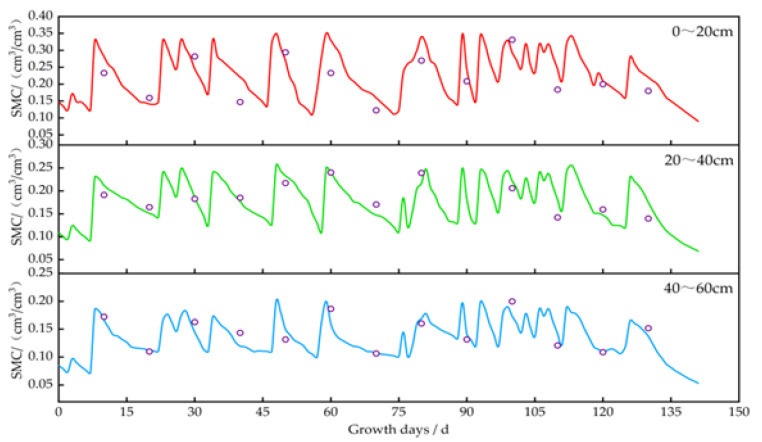
Changes in soil moisture content at a depth of 0–60 cm in potato plots. Note: the circles represent the measured values, and the lines represent the simulated values.

**Figure 7 plants-13-01916-f007:**
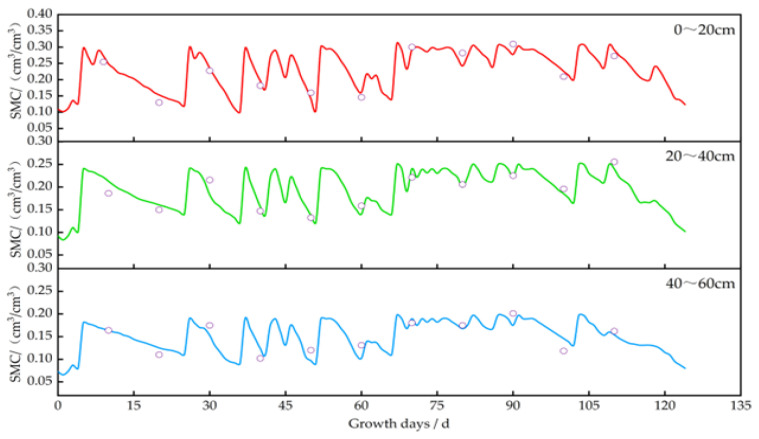
Changes in soil moisture content at a depth of 0–60 cm in oat plots. Note: the circles represent the measured values, and the lines represent the simulated values.

**Figure 8 plants-13-01916-f008:**
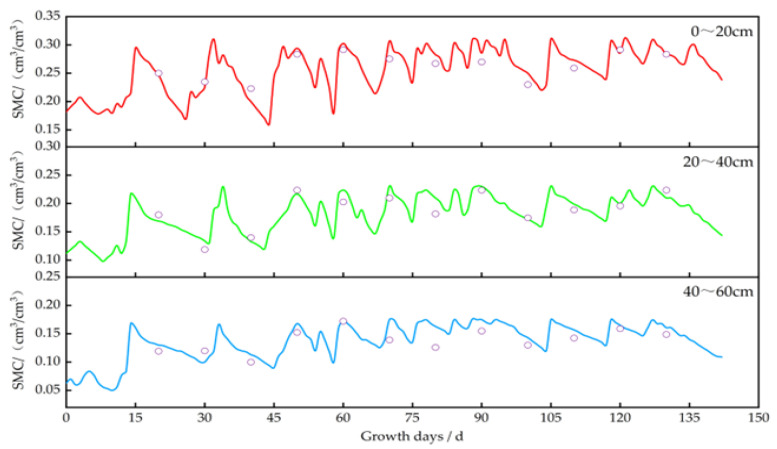
Changes in soil moisture content at a depth of 0–60 cm in alfalfa plots. Note: the circles represent the measured values, and the lines represent the simulated values.

**Figure 9 plants-13-01916-f009:**
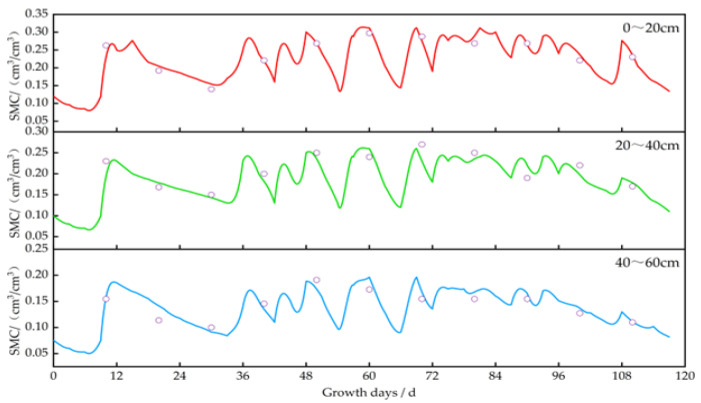
Changes in soil moisture content at a depth of 0–60 cm in sunflower plots. Note: the circles represent the measured values, and the lines represent the simulated values.

**Figure 10 plants-13-01916-f010:**
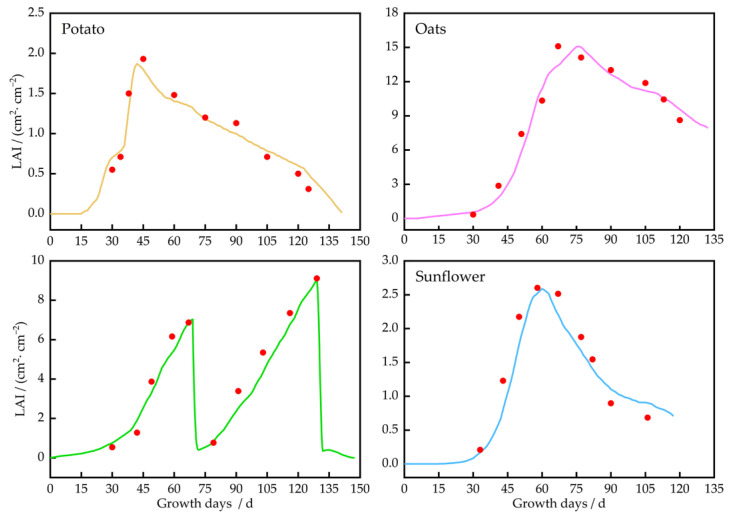
Change trends in leaf area of potatoes, oats, alfalfa, and sunflower. Note: Red dots are measured values.

**Figure 11 plants-13-01916-f011:**
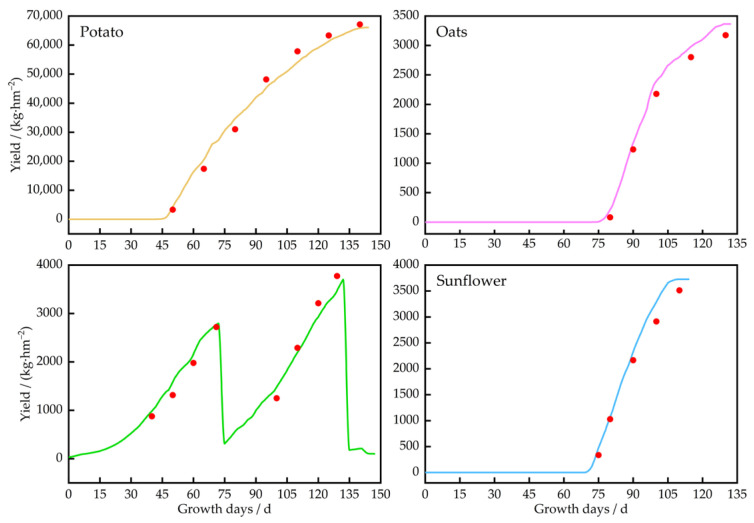
Yield trends in potatoes, oats, alfalfa, and sunflowers. Note: Red dots are measured values.

**Table 1 plants-13-01916-t001:** Basic physical and chemical parameters of *potato* soil.

Soil Layer	Bulk Density		Field Moisture Capacity		Saturated Moisture		Available P		Available K		Soil Organic Matter		pH	
Depth	(g/cm^3^)		(cm^3^/cm^3^)		(cm^3^/cm^3^)		(mg·kg^−1^)		(mg·kg^−1^)		(g·kg^−1^)			
(cm)	2022	2023	2022	2023	2022	2023	2022	2023	2022	2023	2022	2023	2022	2023
0–20	1.34	1.36	0.32	0.30	0.35	0.35	11.59	12.08	163.88	158.72	2.17	2.31	7.29	7.16
20–40	1.57	1.51	0.25	0.24	0.29	0.26	4.56	5.34	132.75	128.31	5.61	5.74	7.11	6.97
40–60	1.58	1.57	0.23	0.18	0.23	0.20	3.09	2.98	74.56	71.78	1.49	1.73	7.08	7.03
60–80	1.62	1.61	0.17	0.21	0.21	0.21	1.84	1.85	66.37	67.23	0.69	0.72	6.96	6.92
80–100	1.65	1.63	0.13	0.14	0.17	0.16	1.57	1.69	92.32	88.75	0.52	0.48	6.93	6.82

**Table 2 plants-13-01916-t002:** Basic physical and chemical parameters of *oat* soil.

Soil Layer	Bulk Density		Field Moisture Capacity		Saturated Moisture		Available P		Available K		Soil Organic Matter		pH	
Depth	(g/cm^3^)		(cm^3^/cm^3^)		(cm^3^/cm^3^)		(mg·kg^−1^)		(mg·kg^−1^)		(g·kg^−1^)			
(cm)	2022	2023	2022	2023	2022	2023	2022	2023	2022	2023	2022	2023	2022	2023
0–20	1.37	1.39	0.29	0.28	0.34	0.33	11.13	11.60	155.69	150.78	2.20	2.34	7.10	7.01
20–40	1.54	1.54	0.23	0.22	0.27	0.24	4.38	5.13	126.11	121.89	5.68	5.81	6.95	6.79
40–60	1.57	1.59	0.21	0.17	0.22	0.19	2.97	2.86	70.83	68.19	1.51	1.75	6.91	6.81
60–80	1.62	1.63	0.16	0.19	0.20	0.20	1.77	1.78	63.05	63.87	0.70	0.73	6.77	6.73
80–100	1.64	1.65	0.12	0.13	0.16	0.15	1.51	1.62	87.70	84.31	0.53	0.49	6.74	6.63

**Table 3 plants-13-01916-t003:** Basic physical and chemical parameters of *sunflower* soil.

Soil Layer	Bulk Density		Field Moisture Capacity		Saturated Moisture		Available P		Available K		Soil Organic Matter		pH	
Depth	(g/cm^3^)		(cm^3^/cm^3^)		(cm^3^/cm^3^)		(mg·kg^−1^)		(mg·kg^−1^)		(g·kg^−1^)			
(cm)	2022	2023	2022	2023	2022	2023	2022	2023	2022	2023	2022	2023	2022	2023
0–20	1.38	1.33	0.28	0.27	0.33	0.31	10.68	11.13	155.69	150.78	2.08	2.14	7.13	7.04
20–40	1.58	1.56	0.22	0.21	0.25	0.22	4.20	4.92	126.11	121.89	5.27	5.40	7.02	6.86
40–60	1.54	1.56	0.20	0.16	0.21	0.17	2.85	2.75	70.83	68.19	1.40	1.63	6.96	6.87
60–80	1.64	1.62	0.15	0.18	0.19	0.19	1.70	1.70	63.05	63.87	0.65	0.68	6.84	6.79
80–100	1.65	1.64	0.11	0.12	0.15	0.14	1.45	1.56	87.70	84.31	0.48	0.42	6.83	6.71

**Table 4 plants-13-01916-t004:** Basic physical and chemical parameters of *alfalfa* soil.

Soil Layer	Bulk Density		Field Moisture Capacity		Saturated Moisture		Available P		Available K		Soil Organic Matter		pH	
Depth	(g/cm^3^)		(cm^3^/cm^3^)		(cm^3^/cm^3^)		(mg·kg^−1^)		(mg·kg^−1^)		(g·kg^−1^)			
(cm)	2022	2023	2022	2023	2022	2023	2022	2023	2022	2023	2022	2023	2022	2023
0–20	1.36	1.31	0.29	0.28	0.31	0.32	11.16	11.63	159.95	154.91	2.19	2.34	7.21	7.02
20–40	1.57	1.54	0.23	0.22	0.27	0.25	4.39	5.14	129.56	125.23	5.54	5.68	7.03	6.84
40–60	1.59	1.55	0.21	0.17	0.22	0.18	2.98	2.87	72.77	70.06	1.42	1.64	6.96	6.88
60–80	1.63	1.61	0.16	0.19	0.20	0.20	1.77	1.78	64.78	65.62	0.67	0.71	6.86	6.76
80–100	1.65	1.63	0.12	0.13	0.16	0.15	1.51	1.63	90.10	86.62	0.49	0.45	6.81	6.76

**Table 5 plants-13-01916-t005:** Growth stages of different crops.

Crop Varieties	Growth Period	2022		2023	
		Start and End Date	Fertility Days	Start and End Date	Fertility Days
*Oat*	Sowing–emergence	19 May~28 May	10 d	21 May~2 Jun.	12 d
	Emergence–jointing	29 May~12 Jul.	45 d	3 Jun.~14 Jul.	41 d
	Jointing–grouting	13 Jul.~6 Aug.	25 d	15 Jul.~8 Aug.	25 d
	Grout–mature	7 Aug.~16 Sep.	40 d	9 Aug.~19 Sep.	41 d
	Total	19 May~16 Sep.	120 d	21 May~19 Sep.	119 d
*Potato*	Sowing–emergence	2 May~27 May	25 d	1 May~28 May	29 d
	Emergence–formation	28 May~26 Jun.	30 d	29 May~29 Jun.	32 d
	Form–expand	27 Jun.~15 Aug.	50 d	30 Jun.~19 Aug.	51 d
	Expansion–harvest	16 Aug.~11 Sep.	25 d	31 Aug.~20 Sep.	21 d
	Total	2 May~11 Sep.	133 d	1 May~13 Sep.	136 d
*Sunflower*	Sowing–emergence	29 May~3 Jul.	36 d	30 May~6 Jul.	38 d
	Emergence–budding	4 Jul.~25 Jul.	21 d	7 Jul.~27 Jul.	20 d
	Budding–blooming	26 Jul.~13 Aug.	18 d	28 Jul.~17 Aug.	20 d
	Bloom–ripen	14 Aug.~26 Sep.	43 d	18 Aug~29 Sep.	41 d
	Total	29 May~26 Sep.	118 d	30 May~28 Sep.	119 d
*Alfalfa*	Greening–branching	11 May~30 May	19 d	10 May~25 May	15 d
	Branching–budding	31 May~22 Jun.	23 d	26 May~18 Jun.	24 d
	Budding–blooming	23 Jun.~10 Jul.	17 d	19 Jun.~5 Jul.	16 d
	Blossom–harvest	11 Jul.~24 Jul.	13 d	6 Jul.~21 Jul.	15 d
	Greening–branching	25 Jul.~10 Aug.	16 d	22 Jul.~4 Aug.	14 d
	Branching–budding	11 Aug.~28 Aug.	17 d	5 Aug.~24 Aug.	19 d
	Budding–blooming	29 Aug.~9 Sep.	12 d	25 Aug.~6 Sep.	12 d
	Blossom–harvest	10 Sep.~20 Sep.	10 d	7 Sep.~18 Sep.	11 d
	Total	11 May~20 Sep.	127 d	15 May~18 Sep.	126 d

**Table 6 plants-13-01916-t006:** Irrigation quotas for different crop growth stages.

Crop Varieties	Growth Period	Irrigation Amount (mm)	
		2022	2023
*Oat*	Sowing–emergence	18	15
	Emergence–jointing	41	36
	Jointing–grouting	52	45
	Grout–mature	35	32
	Total	146	128
*Potato*	Sowing–emergence	32	31
	Emergence–formation	55	50
	Form–expand	66	57
	Expansion–harvest	20	18
	Total	173	156
*Sunflower*	Sowing–emergence	36	23
	Emergence–budding	30	35
	Budding–blooming	46	39
	Bloom–ripen	25	21
	Total	137	118
*Alfalfa*	Greening–branching	16	16
	Branching–budding	39	35
	Budding–blooming	19	16
	Blossom–harvest	18	15
	Greening–branching	19	17
	Branching–budding	38	33
	Budding–blooming	18	18
	Blossom–harvest	15	14.0
	Total	182	164

**Table 7 plants-13-01916-t007:** Evaluation of accuracy indexes for crop leaf area and crop yield.

Crop Type	Statistical Indicators	LAI/(cm^2^·cm^−2^)	Yield/(kg/ha)
*Oat*	ARE/%	3.06	5.82
	nRMSE/%	6.46	5.25
	R^2^	0.87	0.88
*Potato*	ARE/%	4.18	4.89
	nRMSE/%	5.64	6.57
	R^2^	0.88	0.87
*Alfalfa*	ARE/%	3.82	4.27
	nRMSE/%	6.31	5.88
	R^2^	0.87	0.89
*Sunflower*	ARE/%	3.39	3.47
	nRMSE/%	5.32	4.51
	R^2^	0.89	0.90

**Table 8 plants-13-01916-t008:** Evaluation of accuracy indexes for the soil water content.

Crop Type	Soil Depth	ARE/%	nRMSE/%	R^2^
*Oat*	0–20	4.53	8.41	0.86
	20–40	3.96	7.65	0.87
	40–60	5.39	9.73	0.85
*Potato*	0–20	5.27	7.87	0.87
	20–40	6.34	8.65	0.86
	40–60	4.88	6.82	0.88
*Alfalfa*	0–20	4.38	6.22	0.87
	20–40	4.94	5.87	0.87
	40–60	5.57	8.06	0.86
*Sunflower*	0–20	6.36	7.58	0.86
	20–40	3.31	6.34	0.88
	40–60	5.48	6.97	0.87

**Table 9 plants-13-01916-t009:** Genetic parameters of potato varieties.

Argument	Definition	Calibrated Value
*G2*	Leaf area expansion rate [cm^2^·(m^2^·d)^−1^]	1100
*G3*	Potential tuber growth rate [g·(plant·d)^−1^]	23.3
*PD*	Tuber growth stress coefficient (%)	0.9
*P2*	Photoperiod coefficient (%)	0.5
*TC*	Upper limit critical temperature for tubers to start growing	20

**Table 10 plants-13-01916-t010:** Genetic parameters of oat varieties.

Argument	Definition	Calibrated Value
*PIV*	Number of days required for vernalization under optimal temperature conditions (d)	20
*PID*	Photoperiod coefficient (%)	30.5
*P5*	Accumulated temperature during grain filling stage (°C d)	450
*G1*	Number of grains per unit canopy mass of a single plant at flowering stage (grain·g^−1^)	16
*G2*	Standard grain weight under optimal conditions (mg)	24
*G3*	Standard dry mass of stem and spike per plant during the maturation stage under non-stress conditions (g)	1.9
*PHINT*	Accumulated temperature required to complete the growth of one leaf (leaf thermal time) (°C d)	99

**Table 11 plants-13-01916-t011:** Genetic parameters of alfalfa varieties.

Argument	Definition	Calibrated Value
*CSDL*	Critical short day duration (h)	10.5
*PPSEN*	Relative response slope to photoperiod (1/h)	0.2
*EM-FL*	Duration of light and heat from seedling emergence to first blossom appearance (d)	21.5
*FL-SH*	From the initial inflorescence blossoming to the first inflorescence fruit setting, light and heat conditions (d)	6.7
*FL-SD*	The light and heat time from the first inflorescence blooming to the first inflorescence grain production (d)	12.6
*SD-PM*	Photothermal duration from seed production to the first inflorescence’s physiological ripening (d)	33.5
*FL-LF*	The photothermal time between the flowering of the first inflorescence and the cessation of leaf expansion (d)	16
*LFMAX*	Maximum photosynthetic rate of leaves (mg CO_2_/m^2·^s^−1^)	2.5
*SLAVR*	Specific leaf area (cm^2^/g)	290
*SIZLF*	Maximum blade size (cm^2^)	5

**Table 12 plants-13-01916-t012:** Genetic parameters of sunflower varieties.

Argument	Definition	Calibrated Value
*CSDL*	Critical short day duration (h)	15
*PPSEN*	Relative response slope to photoperiod (1/h)	−0.086
*EM-FL*	Duration of light and heat from seedling emergence to first blossom appearance (d)	22.6
*FL-SH*	From the initial inflorescence blossoming to the first inflorescence fruit setting, light and heat conditions (d)	7.2
*FL-SD*	The light and heat time from the first inflorescence blooming to the first inflorescence grain production (d)	12.5
*SD-PM*	Photothermal duration from seed production to the first inflorescence’s physiological ripening (d)	32.5
*FL-LF*	The photothermal time between the flowering of the first inflorescence and the cessation of leaf expansion (d)	15
*LFMAX*	Maximum photosynthetic rate of leaves (mg CO_2_/m^2^·s^−1^)	2.1
*SLAVR*	Specific leaf area (cm^2^/g)	240
*SIZLF*	Maximum blade size (cm^2^)	200
*XFRT*	Maximum proportion of daily dry matter allocated to fruits (g)	0.81
*WTPSD*	Maximum weight per seed (g)	0.1
*SFDUR*	Maximum weight per seed (d)	24
*PODUR*	Optimal photothermal time required for final fruit load (d)	4.5
*THRSH*	Shattering percentage. Maximum ratio of seeds to (seeds + hulls)	72.5
*SDPRO*	Protein content in seeds (g (protein)/g (seed))	0.14
*SDLIP*	Oil content in seeds (g (oil)/g (seed))	0.45

**Table 13 plants-13-01916-t013:** Water balance in the 0–60 cm layer of soil for different crops in 2022 and 2023.

Crop Type	P/mm		I/mm		ΔW/mm		ET/mm		E/mm		T/mm		D/mm	
	2022	2023	2022	2023	2022	2023	2022	2023	2022	2023	2022	2023	2022	2023
*Oat*	191.7	241.1	146	128	−16.5	−18.7	333.8	369.2	73.4	67.1	260.4	302.1	20.4	18.6
*Potato*	185.3	236.0	173	156	−41.6	−44.9	375.2	414.2	254.7	238.0	120.5	176.2	24.7	22.7
*Alfalfa*	213.6	257.8	182	164	−47.4	−56.7	415.7	453.7	184.3	181.2	231.4	272.5	27.3	24.8
*Sunflower*	198.7	241.2	137	118	−37.8	−42.3	355	385.6	160.7	179.6	194.3	206.0	18.5	15.9

**Table 14 plants-13-01916-t014:** Water use rates and irrigation water use rates of different crops in 2022 and 2023.

Crop Type	I/ mm		ET/ mm		Y/ (kg/ha)		Y_a_/ (kg/ha)		WUE/ (kg/m^3^)		IWUE/ (kg/m^3^)	
	2022	2023	2022	2023	2022	2023	2022	2023	2022	2023	2022	2023
*Oat*	146	128	333.8	369.2	3120	3345	2184	2676	1.04	1.01	0.64	0.52
*Potato*	173	156	375.2	414.2	62,355	67,170	43,649	53,736	16.62	16.22	10.81	8.61
*Alfalfa*	182	164	415.7	453.7	4879	6529	3185	5362	1.17	1.44	0.93	0.71
*Sunflower*	137	118	355	385.6	3870	4020	2709	3216	1.39	1.34	0.85	0.68

## Data Availability

Data are contained within the article.

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
