# Peer review of "Exploring the Water–Soil–Crop Dynamic Process and Water Use Efficiency of Typical Irrigation Units in the Agro-Pastoral Ecotone of Northern China"

_plants, 2024, doi:10.3390/plants13141916_

Round 1
Reviewer 1 Report (New Reviewer)
Comments and Suggestions for Authors
In this work, the DSSAT model has been used to simulate the water balance of different crops in the central part of the Inner Mongolia region (China). The model is calibrated with soil moisture measurements and measured crop parameters such as LAI and yield. From the calibrated model, the authors evaluate the drainage water losses and obtain the WUE and IWUE, identifying which crops are adequately irrigated, and which need improvements in irrigation management. Although this is a specific problem in a certain region, the topic of the manuscript is of interest due to the application of the results in the improvement of irrigation efficiency and the adequate management of water resources in the area, and it falls within the scope of the Plants journal. The manuscript needs some improvements. Specific comments and suggestions to the authors are included to improve the final version of the manuscript.
General comments:
- My main objection to the paper is that nothing is explained about model calibration and model parameterization. What method do the authors use? Required inputs, number of simulations, initialization of model runs,… A diagram of the calibration process would be required.
Specific remarks:
- Line 31: it is usual to express crop yield in kg/ha or t/ha.
- Lines 32 – 34: it would be convenient to separate evaporation and transpiration data, so that they can be read more clearly.
- Line 71: ETc
- Lines 112 – 113: but the effective use of the model depends on the results of the validation in each zone and with the crops of that zone.
- Line 134: it is usual to express in ha. In any case, follow the instructions of the journal.
- Line 136: cm/a ¿units? cm/year?
- Line 140: what is the soil texture?
- Lines 165 – 167: How many sensors were installed in each growing area, where were they placed?
The results do not show the values of these sensors. Were the sensor measurements used?
- Table 1: bulk density for 40-60 soil layer depth 2023 is an anomalous data.
- Line 173: Does the field capacity appear in the table? Do you mean the suction pressure shown in the tables 1-4?
- Lines 208 – 209: inputs and outputs of the system
- Line 255: 0.25 and 0.5, respectively.
- Line 259 and 262: check the meaning of Ws (Ws is the solar declination angle (°) in line 259; Ws is the sunset hour angle (°) in line 262).
- Line 265: change Delta to d
- Line 271: physicochemical
- Line 279: What measurements were used for calibration?
- Line 290: Include the units in which you express WUE
- Line 299: How do you measure Ya? Was there a plot without irrigation?
- Lines 310 – 312: Are the parameters listed in Table 9-12 all the genetic crop parameters of each model, or are they the ones selected for calibration?
- Line 334: leaf area index
- Figures 6-9: Indicate in the figure caption that the circles represent the measurements and the lines the simulated values.
- Table 13: The second and third columns represent precipitation, P; the last two columns represent drainage, D.
- Line 453: According to table 13, the precipitation data you mention here correspond to those of alfalfa....
- Line 483-484: water consumption
- Line 507: please check this 415.7.9
- Line 526: Conclusions section. It is not necessary to repeat all the results. They should summarize the main findings of the work done and the applications they have.
Author Response
Dear Professor:
Thank you for taking time out of your busy schedule to review the manuscript. Now we
have carefully corrected and replied to the manuscript for this revision.
Comments 1: My main objection to the paper is that nothing is explained about model calibration and model parameterization. What method do the authors use? Required inputs, number of simulations, initialization of model runs,… A diagram of the calibration process would be required.
|
||||||||||||||||||||||||||||||||||||||||||||||||||||||||||||||||||||||||||||||||||||||||||||||||||||||||||||||||||||||||||||||||||||||||||||||||||||||||||||||||||||||||||||||||||||||||||||||||||||||||||||||||||||||||||||||||||||||
Response 1: Thank you for pointing this out. The explanatory model calibration and model parameterization are reflected in Sections 3.1-3.2 of the article. Calibration and validation are shown below. 2.5.4. Model Calibration and Validation In this study, the absolute relative error (ARE), root mean square error (RMSE), normalized root mean square error (nRMSE), and coefficient of determination (R2) [33, 34,35] were used to measure the relative difference between simulated values from the DSSAT model and actual measurements. Generally, the closer the ARE is to 0, the higher is the model’s accuracy. An nRMSE < 10% is considered excellent; 10–20%, good. The R2 is used to test the fit between model simulations and actual measurements, with values closer to 1 indicating a higher degree of fit. The formula for the index calculation is as follows: (11) (12) (13) (14) Where Mi is the observed value, Si is the simulated value, M is the average of the measured values, S is the average of the simulated values, and n is the number of samples.
3.1. Model Calibration In this study, calibration was conducted using experimental data collected in 2022, with ARE, nRMSE, and R2 serving as the metrics. According to Table 7, the ARE for different crop leaf area indices ranged from 3.06% to 4.18%; the nRMSE values ranged between 5.32% and 6.46%; and the R2 values ranged from 0.87 to 0.89. The ARE for yield ranged from 3.47% to 5.82%, with nRMSE values ranging between 4.51% and 5.88% and R2 values ranging from 0.87 to 0.90. Table 8 shows that the accuracy for soil moisture content at 0–60 cm were ARE ≤7%, nRMSE ≤10%, and R2 ≥ 0.85. The calibration of crop genetic parameters is shown in Tables 9–12. The measured values were good agreement with the simulated values. Table 7. Evaluation of accuracy indexes for crop leaf area and crop yield.
Table 8. Evaluation of accuracy indexes for the soil water content.
Table 9. Genetic parameters of potato varieties.
Table 10. Genetic parameters of oat varieties.
Table 11. Genetic parameters of alfalfa varieties
Table 12. Genetic parameters of sunflower varieties.
3.2. Model Validation After calibration based on the 2022 data, the corrected genetic parameters for different crops (Tables 9–12) were added to the corresponding modules for validation. The model was then validated based on the actual data from 2023. The validation results, as shown in Figure 5, indicated that the ARE for soil moisture in different crops ranged from 4.23% to 6.04%. Meanwhile, the nRMSE values ranged from 4.97% to 5.74% and R2 ranged from 0.901 to 0.912. The LAI in different crops were as follows: ARE, 2.85% to 5.37%; nRMSE, 4.01% to 5.62%, and R2, 0.911 to 0.919. The yields were as follows: ARE, 3.44% to 5.09%; nRMSE, 4.29% to 6.05%; and R2, 0.917 to 0.930. Hence, the model demonstrated good simulation accuracy, and the calibrated and validated models were deemed suitable for use in this study.
Figure 5. Verification of the moisture levels, leaf area index, and yield in different crops.
|
||||||||||||||||||||||||||||||||||||||||||||||||||||||||||||||||||||||||||||||||||||||||||||||||||||||||||||||||||||||||||||||||||||||||||||||||||||||||||||||||||||||||||||||||||||||||||||||||||||||||||||||||||||||||||||||||||||||
Comments 2: Line 31: it is usual to express crop yield in kg/ha or t/ha.
|
||||||||||||||||||||||||||||||||||||||||||||||||||||||||||||||||||||||||||||||||||||||||||||||||||||||||||||||||||||||||||||||||||||||||||||||||||||||||||||||||||||||||||||||||||||||||||||||||||||||||||||||||||||||||||||||||||||||
Response 2: thanks professor giving me useful comments, the whole text has been corrected. Please check it
Comments 3: Lines 32 – 34: it would be convenient to separate evaporation and transpiration data, so that they can be read more clearly.
Response 3: thanks professor giving me useful comments. I have revised it into: soil evaporation of oat, potato, alfalfa, sunflower accounted for 18%~22%, 78 %- 82%; 57%-68%, 32%-43%, and transpiration accounted for 40%-44%, 56%-60%, 45%-47%, 53 %-55% of ET (333.8mm–369.2mm, 375.2mm–414.2mm, 415.7mm–453.7mm, and 355.0mm–385.6mm), respectively. Please check it.
Comments 4: Line 71: ETc.
Response 4: Thanks professor giving me useful comments. The text has been corrected and the changes are shown in the text.
Comments 5: Lines 112 – 113: but the effective use of the model depends on the results of the validation in each zone and with the crops of that zone.
Response 5: Thank you for pointing this out. The text has been added and the changes are as follows: Together, these studies demonstrate that the DSSAT model can effectively evaluate agricultural production and help in enhancing the yield of various crops, but the effective use of the model depends on the results of the validation in each zone and with the crops of that zone. However, current research on typical crop irrigation methods and planting patterns in agropastoral zones mainly relies on field experiments, and the application of crop growth models in this context has thus far been limited.
Comments 6: Line 134: it is usual to express in ha. In any case, follow the instructions of the journal.
Response 6: Thank you for pointing this out. The text has been corrected. Please check it.
Comments 7: cm/a ?units? cm/year? Response 7: Thank you for pointing this out. The unit is cm/a and the cited literature is as follows: Xu, Y. Impact analysis of land use on groundwater level drawdown: a case study of the Hebei Plain[J]. Geographical Research, 2005, 24(2): 222-228.
Comments 8: Why from such a deep layer?Incase of such a heavy textured soil (1.5-1.7 g/cm3 bulk density),is thereany impact of drip irrigation on the soil at 100 cm? only 100 cm was mentioned aboveclarify!
Response 8: Thanks professor giving me useful comments Alfalfa is a perennial pasture with a long main root (100cm during the third year) and a well-developed root system. In order to grow, alfalfa root will uptake the water from different soil layer irrigated by drip irrigation. In addition to taking leakage into account, this study needs to consider the 100 cm soil layer. Thus, the depth of 100cm soil layer was selected to study.
Comments 9: Line 140: what is the soil texture?
Response 9: Thank you for pointing this out. Soil texture is sandy loam. I have added it in my manuscript. Please check it.
Comments 10: Lines 165 – 167: How many sensors were installed in each growing area, where were they placed? The results do not show the values of these sensors. Were the sensor measurements used?
Response 10: According to reviewer’s suggestions, the author has made specific modifications as follows: There were four soil water sensors to installed in study area. Each soil moisture monitoring sensor was installed in each crop field, as shown in figure below.
when we designed experiment, we consider the harsh meteorological conditions in study area. The monitoring equipment we purchased must be suitable for cold areas. Through the comparison and selection of monitoring equipment, we finally decided to purchase moisture monitoring equipment from American METER Company. Since the Chinese agent company has not equipment and can only purchase from the United States, which costs a lot of time to obtian monitoring equipment from American METER Company. Therefore, it was late to install soil moisture monitoring equipment. It was on August 15, 2022 to install soil moisture monitoring equipment. However, in order to make sure the conduction of the experiment, we conducted manual sampling. During the growth period, we took samples every ten days, and each sample repeated three times. In order to ensure the uniform of the soil water data, we uniformly use the water content data measured by manual sampling to calibrate the model. Therefore, the soil water data was not a continuous curve. The measured values was used to validate model, as shown in figure below.
|
Comments 11: Table 1: bulk density for 40-60 soil layer depth 2023 is an anomalous data.
Response 11: Thank you for pointing this out. Data errors have been corrected.
Comments 12: Line 173: Does the field capacity appear in the table? Do you mean the suction pressure shown in the tables 1-4?
Response 12: Thank you for pointing this out. The second column of data in Table 1-4 shows the bulk density, and the suction pressure has been changed to Field moisture capacity.
Comments 13: Lines 208 – 209: inputs and outputs of the system.
Response 13: Thanks professor giving me useful comments. Taking the potato as an example. Crop transpiration, soil evaporation, surface runoff, and soil profile drainage make up the outputs of the system, whereas rainfall and irrigation water was the inputs of the system.
Figure 4. Schematic diagram of water movement process in DSSAT model.
Comments 14: Line 255: 0.25 and 0.5, respectively.
Response 14: Thank you for pointing this out. Changed in the text, as reflected in the text.
Comments 15: Line 259 and 262: check the meaning of Ws (Ws is the solar declination angle (°) in line 259; Ws is the sunset hour angle (°) in line 262).
Response 15: Thank you for pointing this out. I have revised it in my manuscript. Please check it.
Comments 16: Line 265: change Delta to d.
Response 16: Thank you for pointing this out. I have revised it in my manuscript. Please check it.
Comments 17: Line 271: physicochemical.
Response 17: Thank you for pointing this out. I have revised it in my manuscript. Please check it.
Comments 18: Line 279: What measurements were used for calibration?
Response 18: Thank you for pointing this out. Using numerical methods the absolute relative error (ARE), root mean square error (RMSE), normalized root mean square error (nRMSE), and coefficient of determination ( R2) for calibration.
In this study, the absolute relative error (ARE), root mean square error (RMSE), normalized root mean square error (nRMSE), and coefficient of determination (R2) [33, 34,35] were used to measure the relative difference between simulated values from the DSSAT model and actual measurements. Generally, the closer the ARE is to 0, the higher is the model’s accuracy. An nRMSE < 10% is considered excellent; 10–20%, good. The R2 is used to test the fit between model simulations and actual measurements, with values closer to 1 indicating a higher degree of fit. The formula for the index calculation is as follows:
(11)
(12)
(13)
(14)
Where Mi is the observed value, Si is the simulated value, M is the average of the measured values, S is the average of the simulated values, and n is the number of samples.
Comments 19: Line 290: Include the units in which you express WUE.
Response 19: Thank you for pointing this out. I have changed it into: This paper uses water use efficiency (WUE, kg/m3) and irrigation water use efficiency (IWUE, kg/m3) as two indices [36,37] to evaluate and reflect the relationship between crop yield and irrigation water use.
Comments 20: Line 299: How do you measure Ya? Was there a plot without irrigation?
Response 20: Thank you for pointing this out. Ya follows the method of measuring Y to ensure the same methodology and reduce variables; there is not a completely unirrigated planting area. We set the different the patter to simulate the results we want to get. When we get the calibrated model, we do not input the irrigation data (only input meteorological data) to simulated the yield (Ya rainfed yield).
Comments 21: Lines 310 – 312: Are the parameters listed in Table 9-12 all the genetic crop parameters of each model, or are they the ones selected for calibration?
Response 21: Thank you for pointing this out. All the parameters listed in Table 9-12 need to be validated. And all the parameters listed in Table 9-12 have been calibrated.
Comments 22: Line 334: leaf area index
Response 22: Thank you for pointing this out. Changed in the text, as reflected in the text.
Comments 23: Figures 6-9: Indicate in the figure caption that the circles represent the measurements and the lines the simulated values.
Response 23: Thank you for pointing this out. Notes have been added to the name of Figure 6-9, taking Figure 6 as an example:
Figure 6. Changes in soil moisture content at a depth of 0–60 cm in potato plots.
Note: The circles represent the measured value, and the lines represent the simulated values, the same below
Comments 24: Table 13: The second and third columns represent precipitation, P; the last two columns represent drainage, D.
Response 24: Thank you for pointing this out. Yes, it is right. But R is the rainfall, and P is percolation. Maybe it is not clear. According to reviewer’s comments, I revised R into P, and changed P into D.
Comments 25: Line 453: According to table 13, the precipitation data you mention here correspond to those of alfalfa.
Response 25: Thank you for pointing this out. Yes professor. The precipitation of different crops is statistic according to the time of planting and harvesting, different crops have different planting and harvesting period, therefor, different crop have the different rainfall. Alfalfa is a perennial crop that is harvested several times a year. Alfalfa experienced the longest growth period. So the precipitation data is consistent with the rainfall data of alfalfa.
Comments 26: Line 483-484: water consumption.
Response 26: Thank you for pointing this out. Changed in the text, as reflected in the text.
Comments 27: Line 507: please check this 415.7.9.
Response 27: Thank you for pointing this out. Changed in the text, as reflected in the text.
Comments 28: Line 526: Conclusions section. It is not necessary to repeat all the results. They should summarize the main findings of the work done and the applications they have.
Response 28: Thank you for pointing this out. Modify the following:
The consumption of the soil moisture during the rapid growth stage for the potatoes, oats, alfalfa, sunflower was 7–12%,11–13%,7–10% and 10-13% more than that during the middle, middle, mid-to-late and initial growth period, and the yield was 67,170 kg/ha, 3,345 kg/ha, 6529 kg/ha, and 4,020 kg/ha, respectively. Water consumption of oats, pota-toes, and sunflowers was 333.8mm–369.2mm,375.2mm–414.2mm,355.0mm–385.6mm,respectively. The maximum water consumption of alfalfa was 415.7mm–453.7mm. Among the four crops, alfalfa is the highest water consuming crop. The average percolation of the four crop was 12.5mm–15.9mm. The value of water productivity and ir-rigation water use efficiency for potatoes was maximum, ranging from 16.22 to 16.62 kg/m3 and 8.61 to 10.81 kg/m3, respectively, followed by alfalfa, sunflowers, and oats. It was advised that irrigation water could be appropriately reduced to decrease ineffective water percolation and consumption. For the perspective of water productivity, it was recommended that potatoes could be extensively cultivated, alfalfa planted appropriately, and oats and sunflowers planted less.

Reviewer 2 Report (New Reviewer)
Comments and Suggestions for Authors
This research work aimed to search for indexes to indicate more efficient crops in the use of water in the Northern region of China. Based on the results, it was recommended that potatoes could be extensively cultivated, alfalfa planted appropriately, and oats and sunflowers planted less. However, this recommendation was based only on the physical water productivity (kg/m3), which is not always the most appropriate for decision making. The use of socioeconomic indicators, such as economic water productivity, jobs creation (jobs/m3), and economic land productivity, could also be considered, as demonstrated in other studies related to socio-economic indexes for water use in irrigation. Therefore, this limitation should be considered in the manuscript and this approach should be suggested for future studies.
The manuscript is well written. However, the discussion just repeats the results already presented in the previous item and is basically comparative. This section requires improvement.
Additional comments and corrections follow in the attached file

Comments on the Quality of English LanguageThe manuscript is well written
Author Response
Dear Professor:
Thank you for taking time out of your busy schedule to review the manuscript. Now we
have carefully corrected and replied to the manuscript for this revision.
Comments 1: Text strikethrough: The model can be applied to this study.
|
Response 1: OK thanks Professor i have deleted it in my manuscript.
|
Comments 2: Text strikethrough: kg·hm-2.
|
Response 2: Ok , i have deleted it. and other reviewer suggest that kg·hm-2 changed into kg/ha. I have changed it in my manuscript. please check it.
Comments 3: Biomass? it is not clear.
Response 3: Thanks professor giving me useful advice, in the paper, the yield of potato, sunflower, alfalfa and oats is not the biomass. just like yield of the potato, we think potato is the yield, it is not included the leaf, stem and so on. i discussed with other author, we would like to discuss with you whether the yield in the paper can be keep.
Comments 4: Text Highlighting: yield.
Response 4: Thank you for pointing this out. we suggest to keep it.
Comments 5: Text Highlighting: ET.
Response 5: Thank you for pointing this out. ET changed to ETa
Comments 6: actual evapotranspiration??? It is not clear.
Response 6: Thanks professor give me useful advice, i am sorry about "ET". Yes, it is not clear. i have change it into "ETa" in my manuscript, please check it.
Comments 7: oat, potato, sunflower, and alfalfa.
Response 7: Thanks professor giving me useful advise, i have changed it in my manuscript. please check it.
Comments 8: Text strikethrough: Agro-pastoral ecotone of Northern China.
Response 8: Thanks professor giving me useful advice, i have changed it in my manuscript. please check it.
Comments 9: Text strikethrough: different.
Response 9: thanks professor giving me useful advice, i have changed it in my manuscript. please check it.
Comments 10: Label: potato.
Response 10: Thanks professor giving me useful advice, i have changed it in my manuscript. please check it.
|
Comments 11: Text strikethrough: and.
Response 11: Thanks professor giving me useful advice, i have changed it in my manuscript. please check it.
Comments 12: Label: ;.
Response 12: Thanks professor giving me useful advice, i have changed it in my manuscript. please check it.
Comments 13: Text strikethrough: water use efficiency.
Response 13: Thanks professor giving me useful advice, i have changed it in my manuscript. please check it.
Comments 14: Label: water use efficiency.
Response 14: Thanks professor giving me useful advice, i have changed it in my manuscript. please check it.
Comments 15: Text strikethrough: water balance analysis.
Response 15: Ok, i have deleted it.
Comments 16: crop evapotranspiration.
Response 16: Thanks professor giving me useful advice, i have changed it in my manuscript. please check it.
Comments 17: Text strikethrough: different forage crops.
Response 17: ok professor, i have deleted it in my manuscript.
Comments 18: Text strikethrough: FI.
Response 18: Build embankments around the farmland to form a pond and divert water to flood the ground for irrigation. I uploaded pictures as attachments. please check it.
Comments 19: Label: ??
Response 19: Thanks professor giving me useful advise. Build embankments around the farmland to form a pond and divert water to flood the ground for irrigation. I uploaded pictures as attachments. please check it.
Comments 20: Label: of China???
Response 20: Yes, professor. Sorry professor. It is not clear, i have added it in my manuscript.
Comments 21: Text strikethrough: a
Response 21: annual.
Comments 22: Label: year ??
Response 22: Yes, professor. it is year.
Comments 23: Text strikethrough: artificial.
Response 23: Sorry professor. it is not clear. It should be alfalfa. I have changed it in my manuscript.
Comments 24: Label: ??.
Response 24: It should be alfalfa. I have changed it in my manuscript.
Comments 25: Label: ).
Response 25: Ok Professor, i add the altitude.
Comments 26: Label: add the altitude.
Response 26: Thanks professor giving me useful advice, i have added the altitude in my manuscript. Please check it.
Comments 27: Figure 1.Overview of the study area.
Response 27: Thanks professor giving me useful advice, i have cited figure1 in my manuscuript. Please check it.
Comments 28: Figures should be cited in the previous paragraph. See for other figures and tables
throughout the manuscript.
Response 28: I have revised in my manuscript. please check it.
Comments 29: and chemical....
Response 29: Thanks professor giving me useful advice, i have changed it. Please check it.
Comments 30: physical parameters.
Response 30: Thanks professor giving me useful advice, i have changed it. Please check it.
Comments 31: Text strikethrough: forage.
Response 31: Thanks professor giving me useful advice, i have changed it.
Comments 32: Text strikethrough: in soil moisture.
Response 32: Thanks professor giving me useful advice, i have changed it.
Comments 33: Text Highlighting: for44%.
Response 33: Yes, professor. it is right.
Comments 34: Text Highlighting: Soil evaporation and potato transpiration accounted for 68% and 32 % of ET in 2022 and 57 % and 43% of ET in 2023, respectively.
Response 34: Thanks professor giving me useful advice, i have explained it in below.
Comments 35: Why evaporation was higgher than transpirtation for potato??
Response 35: Because potatoes are planted in ridges (figure 4), the distance between the ridges is relatively large, there is no vegetation coverage, and the surface is exposed year-round, resulting in a large amount of soil evaporation. I uploaded pictures of potato cultivation in the experimental area as attachment. The spacing between potatoes is 90cm, as shown in the picture.
Comments 36: Text Highlighting: 3to.
Response 36: Ok professor, i change it.
Comments 37: Text strikethrough: Forage.
Response 37: Ok, i have deleted it.
Comments 38: The discussion repeats the results already presented in the previous item and is basically comparative. This section requires improvement.
Response 38: Thanks professor giving me useful advice, i have revised it in the manuscript. Please check it. Thanks professor.
- 4. Discussion
Using field trial data from 2022 and 2023, this study investigated the soil water dynamics and the growth processes for potatoes, oats, alfalfa, and sunflowers in the agro-pastoral ecotone. The adjusted model aligned well with the measured data(ARE < 10%, nRMSE/% < 10%, and R2 ≥ 0.85), which provided effective predictions of crop growth trends, consistent with findings from Wang et al.(2022)[39], Shen et al. (2020) [40] and Jiang et al. (2019) [41]. The present study also demonstrated that soil moisture changes differed across the different layers of soil at a depth of 0–60 cm, which corresponds to the active rootzone. These results are in line with those from Gao et al. (2015) [42] and Guo et al. (2020) [43], who investigated soil moisture content across different soil depths under various conditions of land use. They demonstrated that the water consumption in the topsoil layer (0–20 cm) is significantly higher than that in the deeper layer (20–40 cm) when the crops have a higher water demand. Similarly, our study revealed that the reduction in soil water within the surface layer (0–20 cm) was 6%–11% and 15%–19% greater than that in the deeper layer (20–60 cm) for potatoes in 2022 and 2023, 11%–18% and 13%–21% higher for oats, 3%–4% and 7%–11% higher for alfalfa, and 2%–5% and 6%–10% higher for sunflowers. Additionally, the majority of roots in potato and oat plants are distributed at a depth of 0–40 cm [44,45]. The significant differences in water absorption and utilization among the four crops led to variations in leaf growth, with the peak LAI being 1.9 for potatoes; 16 for oats; and 2.7 for sunflowers. Meanwhile, alfalfa exhibited multiple LAI peaks (7.5 and 9). The sensitivity to water deficit across various yield accumulation stages differs among different crops [46], and these differences lead to varying rates of yield increase. For example, Song et al. [47] studied the coupling effect of water and nitrogen on potato yield, quality, and water productivity under subsurface drip irrigation conditions in the northwest arid area of Weiwu City. They found that P2N2 (soil moisture ratio of 70% and nitrogen application rate of 135 kg/ha) treatment was the best performance in potato tuber starch content and yield, with the highest yield of 54187 kg/ha, and P1N2 (soil moisture ratio of 40% and nitrogen application rate of 135 kg/ha) treatment was the highest water productivity of 12.86 kg/m3. This study found that the yield was 62355-67170kg/ha. The potato yield in this study is higher,since the nitrogen fertilizer (N) applied in this study was 185kg/ha, which was 50 kg/ha more than that of Song et al, and the rainfall in Weiwu City was less than 100mm.
Simulations based on the DSSAT model were conducted to track water consumption changes, ET, and yield for potatoes, oats, alfalfa, and sunflowers [48]. The model accurately depicted the water requirements throughout the crop growth process, highlighting the critical role of concentrated rainfall and irrigation in supporting crop growth and enhancing soil moisture content in arid regions [49]. This study identified varying water consumption levels at different stages of crop growth. The total water consumption was found to be 375.2–414.2 mm for potatoes, 333.8–369.2 mm for oats, 415.7–453.7 mm for alfalfa, and 355.0–385.6 mm for sunflowers. The water consumption for oats was consistent with the value reported by Hao et al. (2019) [50], whereas the consumption for potatoes was higher than the total water consumption of 216–278 mm reported by Chen et al. (2019) [51]. Conversely, the water consumption calculated for alfalfa and sunflowers was lower than the previously reported water consumption of 400–500 mm for alfalfa [52] and 400 mm for sunflowers [53] during the growth season. The lower values observed for alfalfa and sunflowers were likely due to factors such as shorter growth seasons, less rainfall, and insufficient heat accumulation. Additionally, the water percolation for potatoes, oats, alfalfa, and sunflowers was 20.1–22.7 mm, 15.9–18.6 mm, 22.7–24.8 mm, and 12.5–15.9 mm, respectively. This suggests that agricultural irrigation management practices should focus on controlling leakage during this period to reduce ineffective water consumption. The irrigation water for the four crops could be reduced to potentially decrease ineffective percolation and consumption. The WUE of alfalfa in 2023 was higher than that in 2022, whereas the WUE for oats, potatoes, and sunflowers all decreased from 2022 to 2023, indicating the superior water utilization capability of alfalfa.
Overall, the optimal planting patterns for crops under different water and fertilizer combinations will be examined in future studies based on the DSSAT model, providing a scientific basis for sustainable agricultural development in arid regions.
Comments 39: Additionally, the majority of roots in potato and oat plants are distributed at a depth of 0–40 cm.
Response 39: I explain it in below. We measured the root length of potatoes and oats and verified it by consulting reference. Upload reference files as attachments. please check it.
1.Boguszewska-Mańkowska D.; Zarzyńska K.; Nosalewicz A. Drought differentially affects root system size and architecture of potato cultivars with differing drought tolerance[J]. American Journal of Potato Research, 2020, 97(1): 54-62.
2.Osborne S L.; Chim B K.; Riedell W E. Root length density of cereal and grain legume crops grown in diverse rotations[J]. Crop Science, 2020, 60(5): 2611-2620.
Comments 40: Where these data??
Response 40: We measured the root length of potatoes and oats and verified it by citing reference. Upload reference files as attachments. please check it.

This manuscript is a resubmission of an earlier submission. The following is a list of the peer review reports and author responses from that submission.
Round 1
Reviewer 1 Report
Comments and Suggestions for Authors
The manuscript is confusing in its present form, and therefore its interest would be greatly limited. My suggestion is a complete revision and an eventual new resubmission. A clearer vision of the objectives should be given in the introduction, that is now too much focalized to local issues. Also, the order by which results, methodology, discussion are presented is disorienting the reader.
Reviewer 2 Report
Comments and Suggestions for Authors
The manuscript provides specific knowledge about the assessment of soil-plant-water systems based on modelling their water regime. The research aimed to provide a theoretical basis for the efficient utilization of agricultural water resources, and rational optimization of cropping patterns. Several practically useful pieces of information are involved in the Results section that are suitable to be the basis of specific conclusions to be drawn.
I still consider a better approach in the justification of the study focussing on these conclusions rather than comparing the water regimes of the different crops under study.
Enough relevant national and international references are cited in the manuscript.
The study is quite complete, except for the methods part.
The titles of the figures and tables are not informative enough. I consider some figures not necessary (indicated in the text).
Several duplications can be found in the tables and the text.
The Conclusions part is missing the specific conclusions included in the results. This part should be harmonized with the part describing the water use efficiency data of the crops under study as these are the most valuable and specific results providing novel knowledge.
Legends and explanations of several abbreviations are missing.
I inserted some sticky notes with my specific comments in the pdf file of the manuscript.
